# Collagen-Based Ovule Therapy Reduces Inflammation and Improve Cervical Epithelialization in Patients with Fungal, Viral, and Bacterial Cervico-Vaginitis

**DOI:** 10.3390/medicina59081490

**Published:** 2023-08-18

**Authors:** Nicoletta De Rosa, Fabrizia Santangelo, Celestino Todisco, Fabiana Dequerquis, Claudio Santangelo

**Affiliations:** 1Ginecology and Obstetric Department, Pio XI Hospital, 20832 Monza-Brianza, Italy; 2Hospital of National Relevance “A. Cardarelli”, 80113 Naples, Italy; fabriziasantangelo@hotmail.it (F.S.); f.dequerquis@libero.it (F.D.); 3ASL Napoli 3 Sud, 80059 Torre del Greco, Italy; c.todisco@aslnapoli3sud.it

**Keywords:** vulvovaginal infections, cervico-vaginitis, cervical epithelialization, vaginal microbiota, human papilloma virus

## Abstract

*Background and Objectives*: Vulvovaginal infections pose significant health challenges for women, necessitating effective treatment approaches. This retrospective observational study aimed at investigating the efficacy of collagen-based vaginal ovules therapy, specifically Plurigin Ovules, in restoring cervical epithelialization and reducing inflammation in mycotic, viral, and bacterial cervico-vaginitis. *Materials and Methods*: A total of 398 women with cervico-vaginitis were included in the study, categorized into three groups: bacterial cervico-vaginitis (Group A), viral cervico-vaginitis (Group B), and fungal cervico-vaginitis (Group C). Participants received vaginal therapy with Plurigin Ovules for three months. Vaginal health parameters were assessed at baseline (T0) and after six months (T1) using various diagnostic tests and analyzed with appropriate statistical tests. *Results*: Significant improvement in cervico-vaginitis was observed in all three groups. At T1, 87.7% patients of Group A, 66.7% of Group B, and 71.5% of Group C achieved infection resolution (all *p* < 0.05). Positive colposcopy results decreased across all groups (*p* < 0.001). Positive vaginal swabs and altered vaginal pH decreased in group A and C (*p* < 0.001). Positive HPV tests decreased in Group B (*p* < 0.001). Positive Pap tests and clinical examinations decreased significantly across all groups (*p* < 0.001). The odds ratios were calculated to reveal the significant associations between these diagnostic outcomes. The therapy was well-tolerated, and no major adverse events were reported. *Conclusion*: Plurigin Ovules exhibited promising therapeutic outcomes in the three cervico-vaginitis conditions studied. Bacterial cervico-vaginitis showed the most significant improvement, followed by fungal and viral cervico-vaginitis. These findings emphasize the potential of Plurigin Ovules as an effective therapeutic option for cervico-vaginal inflammation and infection, highlighting its role in promoting re-epithelialization and reducing inflammatory processes in the cervix and vagina.

## 1. Introduction

Vulvovaginal infections encompass a wide array of conditions affecting the female reproductive tract, which includes the vulva, vagina, and cervix [1]. These infections are a common reason for women to seek medical care, due to discomfort, distress, and a significant burden on the quality of life they cause. Vulvovaginal infections can be categorized into mycotic (fungal), viral, and bacterial infections [2]. Among these, the most prevalent infections encompass vulvovaginal candidiasis, bacterial vaginosis, and sexually transmitted infections like human papillomavirus (HPV) and herpes simplex virus (HSV) [1,2].

Maintaining a balanced vaginal microbiota and an intact cervical epithelium is crucial to ensure vaginal health and overall well-being. The vaginal microbiota comprises a complex ecosystem of microorganisms, predominantly lactobacilli. These essential microorganisms sustain a low vaginal pH, produce antimicrobial compounds, and provide a natural defense mechanism against pathogenic microorganisms [3]. Disturbance in the normal vaginal microbiota can result in dysbiosis, leading to an overgrowth of harmful microorganisms and subsequent infections [4]. Similarly, the cervical epithelium functions as a protective barrier against microbial invasion preserving the vaginal environment’s integrity [5].

Of particular significance in maintaining vulvovaginal health is the transformation zone (TZ) within the cervix [6]. This transitional area, situated between the exocervical squamous epithelium and the endocervical columnar epithelium, undergoes a natural process known as metaplasia [7]. Metaplasia involves the transformation of columnar epithelium into the squamous epithelium [7]. However, this process can be gradual and susceptible to infections caused by viruses such as HPV, fungi, or bacteria. Disruptions in the metaplastic process can lead to chronic inflammation and increase the risk of cervicitis, vaginitis, or endometritis [6,7]. A healthy cervix and vaginal microbiota represent a strong barrier against infection development, thereby reducing the probability of recurrences or persistent infections.

Current therapeutic approaches to vulvovaginal infections often include antimicrobial agents, antifungals, and antivirals [8]. While these treatments can provide symptomatic relief, they often fail to address the underlying imbalances in the vaginal microbiota and to facilitate the restoration of the cervical epithelium. Moreover, the emergence of drug-resistant strains and the potential for adverse effects highlight the need for alternative treatment therapies [9].

In recent years, there has been growing interest in exploring novel therapeutic strategies aimed at restoring the vaginal microbiota and enhancing cervical epithelialization. Among these approaches is vaginal ovum therapy, which involves the administration of specific compounds directly into the vaginal canal to restore microbial balance and promote tissue healing.

Plurigin Ovules is a non-sterile, disposable medical device that promote the healing process in case of inflammatory and dystrophic vaginal mucosal conditions [10]. By combining an acidifying compound (lactic acid), and a bio-adhesive substance that also contains collagen (swine gelatin), this device aims to simultaneously restore vaginal pH, and enhance tissue repair in the vaginal and cervical areas.

Given the challenges posed by vulvovaginal infections and the limitations of current treatment options, the present study’s primary goal is to assess the efficacy of Plurigin Ovules. Specifically, the study aims to evaluate their capacity to promote re-epithelialization and mitigate inflammatory processes within the cervix and vagina. This study was designed to retrospectively evaluate the therapeutic outcome of Plurigin Ovules in three distinct conditions of cervicovaginal inflammation and infection: mycotic, viral, and bacterial. In addressing these diverse conditions, we aim to provide a comprehensive overview of the potential of Plurigin Ovules as a treatment alternative, addressing important gaps in the management of cervico-vaginal infections.

## 2. Materials and Methods

### 2.1. Study Design

This study utilized a retrospective observational design to investigate the restoration of cervical epithelialization and the reduction of inflammatory processes in patients with mycotic, viral, and bacterial cervico-vaginitis, undergoing therapy with Plurigin Ovules, who visited the Gynecology and Obstetrics Outpatient Clinics at A.O.R.N. Cardarelli, Naples, Italy, between 1 January 2018, and 15 April 2021. Clinical cases that met the inclusion criteria were included in this study.

The study was approved by the A.O.R.N. Cardarelli ethics committee, deliberation number 607 of 13 July 2017. All procedures adhered to the ethical standards set forth by the responsible committee on human experimentation (both institutional and national), as well as the Helsinki Declaration of 1975, as revised in 2008.

### 2.2. Study Population

Approximately 500 women were screened. The selection of participants was based on the following inclusion criteria:Age between 18 and 60 years;With a confirmed diagnosis of cervico-vaginitis (as specified in the next section);Reporting symptoms, such as burning, itching, and leukorrhea;Women who underwent a gynecological and instrumental follow-up exams at time 0 (baseline time) and after 6 months (follow-up).Having performed during the follow-up period a therapy with Plurigin Ovules that consisted in 1 application/day for 10 days per month, for 3 months.

Exclusion criteria were applied to ensure a homogeneous study population. The following cases were excluded from the sample:Patients with a diagnosis of cervical intraepithelial neoplasia grade 2 or higher (CIN2+);Individuals with acute cervico-vaginitis, characterized by severe inflammation of the cervix and vagina.Patients who were currently undergoing specific antibiotic or antifungal therapy at the time of the initial visit;Individuals with a history of immunosuppression, such as those with compromised immune function;Patients with dysendocrinopathies, including diabetes or thyroid disorders;Pregnant women.

### 2.3. Data Collection

Each subject included in the study was contacted and provided with detailed information regarding the purpose and procedures of the study. Written informed consent was obtained from each participant prior to inclusion in the study.

Clinical records were searched in the database to identify cases that met the predetermined inclusion criteria. The diagnosis of cervico-vaginitis was extracted from clinical records and defined based on specific criteria, established as indicators of cervico-vaginal health. Diagnostic confirmation required at least two of the following criteria:Pap-test report of: Moderate or severe inflammation and/or atypical squamous cells of undetermined significance (ASCUS) and/or low-grade squamous intraepithelial lesion (LSIL);Colposcopy report of: Colpitis, symptomatic ectropion > 1/3, colposcopic findings of immature metaplasia;Vaginal swab report: Positive for pathogenic microorganisms;Human Papilloma Virus (HPV) test report: Positive for HPV. HPV DNA testing was conducted using the HC-2 assay to detect the presence of HPV.Clinical examination (speculum examination) findings: Leukorrhea (foamy, dense, yellowish), vulvovaginal erythema, edema;Vaginal pH measurement: Altered pH levels.

The identified clinical cases were classified into three groups based on the etiological agent responsible of the infection:Group A: Women with bacterial cervico-vaginitis.Group B: Women with viral cervico-vaginitis.Group C: Women with fungal cervico-vaginitis.

Pertinent information regarding participants’ clinical history, age, lifestyle habits (smoking, alcohol consumption), and sexual activity (number of lifetime sexual partners) were extracted from the clinical records. This data collection process has been designed to ensure the selection of suitable participants and the acquisition of necessary information for the evaluation of study objectives and outcomes. Confidentiality and privacy of participants’ data were strictly maintained throughout the study.

### 2.4. Clinical Investigation Endpoints

The primary endpoint of this study was to evaluate the restoration of cervical epithelialization and the resolution of inflammatory processes in the cervix and vagina, in patients with mycotic, viral, and bacterial cervico-vaginitis undergoing vaginal therapy with Plurigin Ovules.

The safety endpoint focused on recording any adverse events or complications associated with the medical device use, as reported by patients.

### 2.5. Medical Device Description

Plurigin Ovules is a non-sterile, disposable medical device intended to promote reparative processes in inflammatory, infective and dystrophic diseases of the vaginal mucosa [10]. It is also indicated as an adjuvant in healing processes, and as a moisturizer in vaginal dryness. The supportive function in the processes of re-epithelialization and hydration can be attributed to both swine gelatin and lactic acid. Lactic acid contributes to maintaining a low vaginal pH, which inhibits pathogenic organisms. Swine gelatin, which contains collagen, imparts bio-adhesive properties to the ovules, ensuring prolonged contact with the vaginal mucosa, facilitating drug delivery, and enhancing therapeutic efficacy. In this way, the two substances protect the mucosa from the external environment. This favors the physiological restoration of tissues and the normalization of damaged surfaces.

### 2.6. Data Analysis

A power calculation was conducted to determine the sample size required to detect meaningful differences in the effectiveness of Plurigin Ovules therapy across different types of cervico-vaginal infections. The parameters considered for the power calculation included an expected effect size based on preliminary data, a significance level (alpha) of 0.05, and a desired power of 0.80. Given the anticipated variability in treatment response among bacterial, viral, and fungal cervico-vaginitis cases, a sample size of approximately 500 participants was deemed necessary. This sample size ensured that the study had sufficient statistical power to detect significant differences in therapeutic outcomes, allowing for a comprehensive evaluation of the therapy’s effectiveness in promoting cervical homeostasis and reducing infection recurrence. Statistical analysis was performed using the SPSS software package. The significance level was set at *p* < 0.05. The Shapiro-Wilk test was employed to assess the distribution of data. Differences between groups for ordinal variables (e.g., Pap smear/HPV test, colposcopy) and within-group comparisons (baseline vs. 6-month follow-up) were evaluated using the chi-square test. The Mantel-Haenszel test was used to calculate odds ratio (OR) and the correspondent confidence intervals (CI). The odds ratios were interpreted in relation to 1.0. An odds ratio greater than 1.0 indicated a higher likelihood of the event occurring in the exposed group compared to the reference group, while an odds ratio less than 1.0 suggested a lower likelihood. For numerical variables (e.g., age), either the ANOVA or Kruskal-Wallis test was applied, depending on the data distribution.

## 3. Results

### 3.1. Patients

A total of 398 patients who met the inclusion criteria and completed the required follow-up as per the protocol were included in the analysis. Among them, 130 patients was assigned to group A (bacterial infections), 117 patients to group B (viral infections), and 151 patients to group C (fungal infections). The median age in Group A was 32.3 ± 4.9, in Group B was 31.7± 4.2, in Group C was 31.5 ± 4.1. The demographic and baseline characteristics of the study subjects are presented in Table 1. There were no significant differences in demographic and baseline characteristics among the groups. The study population consisted of 70.4% (*n* = 280) non-smokers, 86.7% (*n* = 345) non-daily alcohol consumers, and 56.5% (*n* = 225) individuals who reported having fewer than four sexual partners in their lifetime. These percentages did not differ significantly when analyzed across groups.

### 3.2. Efficacy

Overall, significant improvement in cervico-vaginitis was observed in all three groups at the end of the study period (T1) (Table 2). When stratified by etiological agent, a significantly greater improvement was observed in group A (bacterial cervico-vaginitis) compared to group B (viral cervico-vaginitis) and group C (fungal cervico-vaginitis) (Table 2, *p* < 0.05). Indeed, at 6-month follow-up (T1), 87.7% of women in group A (*n* = 114) achieved resolution of infection, compared to 66.7% in group B (*n* = 78) and 71.5% in group C (*n* = 108) (Table 2, *p* < 0.05).

The odds ratios (OR) were calculated to further elucidate the comparative effectiveness among the groups and are presented in Table 3. In Group A, the odds of achieving infection resolution were significantly higher by a factor when compared to Group B (viral cervico-vaginitis) and Group C (fungal cervico-vaginitis) combined. Specifically, the odds of successful outcomes in Group A were 7.00 times higher than the odds of unsuccessful outcomes. In contrast, in Group B, the odds of achieving infection resolution 2.00 times higher than the odds of unsuccessful outcomes. Similarly, in Group C, the odds of achieving resolution were 2.50 times higher than the odds of unsuccessful outcomes.

Table 4 presents the vaginal health parameters stratified by group, at baseline (T0) and after 6 months (T1). At baseline (T0), group A showed a higher incidence of abnormal Pap test results (*p* < 0.001 vs. Group B and C). This higher incidence could be attributed to the inflammatory response induced by bacterial cervico-vaginitis at the cellular level. A higher incidence of positive HPV tests was observed in both viral cervicitis (group B, *p* < 0.001 vs. group C) and bacterial cervico-vaginitis (group A, *p* < 0.001 vs. group C), possibly due to the coexistence of bacterial vaginosis and viral persistence. However, this association was not observed in fungal infections, suggesting a different pathogenesis. Furthermore, group A exhibited a higher incidence of clinically apparent manifestations, followed by group C and group B. Both group A and group C had a higher incidence of positive vaginal swabs (*p* < 0.001 vs. group A and B). There were no statistically significant differences in colposcopy results among the three groups, as approximately 90% of cases showed evident colposcopic abnormalities.

At 6-month follow-up (T1), there has been a decrease in the number of positive colposcopy results compared to baseline (T0) for all three groups (all *p* < 0.001). However, no significant differences were found in the incidence of positive colposcopy among the groups. Figure 1 displays representative colposcopy images from three different cases. Positive vaginal swabs were decreased in Group A and C (*p* < 0.001) compared to baseline. Group C exhibited a higher occurrence of positive vaginal swabs (*p* < 0.001 vs. Group A and B), indicating a greater recurrence/persistence rate of fungal infections. The percentage of patients with altered vaginal pH decreased in Group A and C (*p* < 0.001 vs. baseline). No differences among groups were observed. Positive HPV tests significantly decreased in Group B (*p* < 0.001 vs. baseline). As expected, HPV recurrence rate was higher in Group B (viral) compared to other Groups (*p* < 0.001 vs. Group A and C). Both positive Pap tests and positive clinical examinations significantly decreased in all 3 groups compared to baseline (*p* < 0.001). Compared to Group B and C, Group A continued to exhibit a higher incidence of positive clinical examination findings (*p* < 0.001 vs. Group B and C).

Case 1—BASELINE: colposcopy view after acetic acid application reveals a normal transformation zone, visible squamocolumnar junction (SCJ), 2/3 ectropion, original trophic epithelium. TREATMENT: colposcopy view after acetic acid application reveals a normal transformation zone, visible SCJ, complete metaplasia with Naboth cysts. Case 2—BASELINE: (A) colposcopy view of abnormal transformation zone grade 1, with extended white epithelium in all four quadrants with geographic borders, visible SCJ; (B) colposcopy view with Lugol’s iodine solution application, reduced uptake; TREATMENT: (A) colposcopy view after acetic acid application reveals a normal transformation zone and an original trophic epithelium; (B) colposcopy view with Lugol’s iodine solution application shows a complete uptake. Case 3—BASELINE: (A) colposcopy view with acetic acid application reveals an abnormal transformation zone grade 1, extended white epithelium in all four quadrants with geographic borders and visible SCJ; (B) colposcopy view with Lugol’s iodine solution application, reduced uptake. TREATMENT: (A) colposcopy view after acetic acid application with a normal transformation zone and an original trophic epithelium; (B) colposcopy view with Lugol’s iodine solution application, complete uptake.

### 3.3. Safety

The therapy was well-tolerated, and no major adverse events were reported in any of the three groups during treatment and follow-up. The treatment’s safety profile was consistent across all groups.

## 4. Discussion

This retrospective observational study demonstrates that topical treatment with Plurigin Ovules promotes cervical-vaginal homeostasis, reducing the recurrence and/or persistence rates of cervical-vaginal infections (bacterial, viral and mycotic), with a greater impact on women with bacterial cervico-vaginitis. The data presented show that colposcopy findings and vaginal examination significantly improved in treated patients. After 6 months follow-up, 75.4% (*n* = 300) of patients were cured and did not experience recurrence. The highest healing rate was observed in bacterial infections, followed by fungal infections, where vaginal homeostasis plays a major role.

The cervical squamous epithelium represents the first and most critical form of protection in the genital environment, providing a mechanical barrier that protects against opportunistic pathogen invasion [11]. In the absence of this barrier, during a period of active cell replication such as that occurring in the presence of immature metaplasia or cervical ectropion, in a state of cellular inflammation, and in the presence of contributing factors such as altered vaginal pH, various pathogens can replicate more easily and disrupt the normal resident microbiota [11,12]. Our results align with previous research highlighting the importance of a balanced vulvo-vaginal environment in maintaining vaginal health [10,13,14,15,16,17]. These findings are consistent with the emerging understanding that preserving a stable vaginal microbiome may be more beneficial than the use of antibiotics or antiviral agents alone in managing various cervico-vaginal infections. The observed reduction in recurrence rates after the 6-month therapy period further substantiates the potential clinical value of Plurigin Ovules in preventing chronic and persistent infections.

Our data show that in patients with cervico-vaginitis, 3-month cyclic therapy allows a significant improvement in colposcopy findings. This is due to maturation and re-epithelialization of squamous epithelium, resolution of cervical ectropion and colpitis in all three study groups. This effect is demonstrated by the incidence of normalized Pap test in all study groups, suggesting a reduction in inflammatory processes, and vaginal swab improvement in Group A and group C. The vaginal device likely helps rebalance the local microbial flora and reducing inflammation, thus preventing contamination by other microbial forms.

Persistent infection with HPV strains is considered an etiological factor responsible for the development of cervical cancer [18]. It has recently emerged that the vaginal microbiome is a significant variable that influences the natural history of HPV infection and the progression of precancerous lesions. The vaginal microbiome and the resident lactobacilli population can positively interfere with the local immune response. Conversely, a vaginal population typical of bacterial vaginosis is involved in cervical tumor oncogenesis and reduces HPV clearance [17].

Our data show significant molecular test improvement in group B (viral infections), with approximately 60% of patients becoming negative for the virus after only 6 months. According to literature, HPV viral clearance is highly variable depending on viral genotype and women’s lifestyle habits, but reaches approximately 50% within 12 months [19]. Even after ablative treatment of the lesion, it settles around 50% [19]. According to our findings, the therapy accelerated the viral clearance process in HPV-positive individuals without precancerous lesions.

The therapeutic potential of ovule-based treatments for cervico-vaginitis, which allows a localized drug delivery, has been the subject of several studies [20,21,22,23]. Collectively, these investigations have highlighted the feasibility and potential of ovule-based therapies, leading to advancements in the management of cervico-vaginal infections. Plurigin Ovules are unique among ovule-based therapies due to the inclusion of collagen (present in the swine gelatine) in the formulation, a distinct feature that distinguishes them from previous interventions. Collagen, a fundamental component of connective tissue, possesses inherent properties known for their potential in wound healing, tissue repair, and immune modulation [24,25]. Exploiting the properties of collagen, Plurigin Ovules have shown to be able to stimulate tissue regeneration and modulate inflammation in cervico-vaginal tissue.

The study has some limitations as it is a retrospective study, which inherently carries the risk of selection bias and limited control over data collection and therefore the data need to be prospectively confirmed. However, the sample size and selective inclusion criteria ensure its strength and validity. Further studies (case-control) are needed to establish the impact of adjuvant therapy on the resolution of different infectious conditions.

Supporting our work, it is now well-known that a balanced vulvo-vaginal environment is crucial for maintaining vaginal health [16]. Clarifying that Plurigin Ovules has a positive effect on every form of cervico-vaginitis, thanks to the improvement of the vaginal environment and cervical re-epithelialization, is of considerable scientific interest.

## 5. Conclusions

In conclusion, this retrospective observational study provides evidence supporting the efficacy and safety of a topical treatment with Plurigin Ovules in maintaining cervical-vaginal homeostasis and reducing cervical-vaginal infections recurrence and persistence. The findings highlight the significant improvements observed in colposcopy findings and vaginal examination following the therapy regimen. The high healing rate and the absence of recurrence after 6 months of treatment demonstrate the therapeutic potential of this approach, particularly in bacterial cervico-vaginitis cases. Our study supports the notion that maintaining a balanced vulvo-vaginal environment is crucial for overall vaginal health. These findings contribute to existing knowledge and encourage further research in prospective studies to validate and explore the potential of this adjuvant therapy. Emphasizing the importance of this approach may lead to a reduction in the excessive use of antibiotics and antifungal therapies. This may promote a more targeted and effective management of diverse infectious conditions.

## Figures and Tables

**Figure 1 medicina-59-01490-f001:**
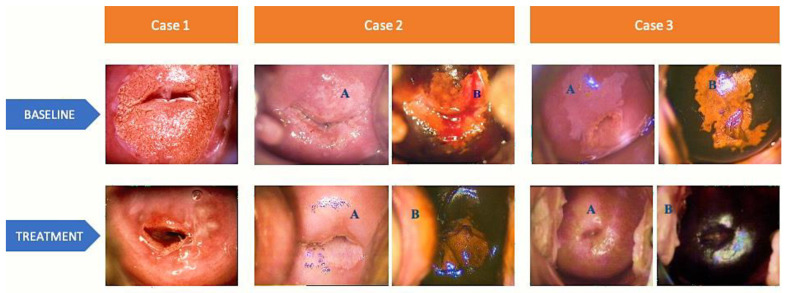
Colposcopy images of 3 representative cases at baseline and after collagen-based ovule therapy. (**A**) colposcopy view after acetic acid application and (**B**) colposcopy view with Lugol’s iodine solution application

**Table 1 medicina-59-01490-t001:** Demographic and anamnestic characteristic of the 3 study groups. Differences between the groups are not significant (NS).

	Group A *n* = 130	Group B *n* = 117	Group C *n* = 151	*p* Value
**Age**	32.3 ± 4.9	31.7 ± 4.2	31.5 ± 4.1	NS
**Parity (range)**	1 (0–3)	1 (0–3)	1 (0–3)	NS
**Smoking habits (%)**				NS
yes	95 (73.1)	73 (62.4)	112 (74.2)	
not	35 (26.9)	44 (37.6)	39 (25.8)	
**Alcohol habits (%)**				NS
yes	120 (92.3)	98 (83.8)	127 (84.1)	
not	10 (7.7)	19 (16.2)	24 (15.9)	
**Sexual partners (%)**				NS
≤4	76 (58.5)	64 (54.7)	85 (56.3)	
>4	54 (41.5)	53 (45.3)	66 (43.7)	

**Table 2 medicina-59-01490-t002:** Incidence of cervico-vaginitis after the follow-up period in the 3 study groups. Negative patients are defined as patients without any sign of cervico-vaginitis, as defined in the Material and Methods section. * *p* < 0.05 compared to baseline (*n*).

Cervico-Vaginitis Profile at the End of the Follow-Up Period		Goup A (*n* = 130)	Group B (*n* = 117)	Group C (*n* = 151)	*p* Value
Number of negative patients	(%)	114 *	78 *	108 *	<0.05
		(87.7)	(66.7)	(71.5)	
Number of positive patients	(%)	16	39	43	
		(12.3)	(33.3)	(28.5)	

**Table 3 medicina-59-01490-t003:** Odds ratios and 95% confidence intervals (CI) for infection resolution among study groups. This table presents the calculated odds ratios (OR) and their corresponding 95% confidence intervals (CI) to assess the comparative effectiveness of infection resolution among different study groups. The odds ratios were computed as the ratio of the odds of infection resolution in one group to the odds of infection resolution in another group.

Comparison	Odds Ratio (OR)	95% Confidence Interval (CI)
Group A vs. Group B	7.00	(3.45, 14.21)
Group A vs. Group C	7.00	(3.63, 13.42)
Group B vs. Group C	1.00	(0.51, 1.96)

**Table 4 medicina-59-01490-t004:** A descriptive table showing vaginal health parameters stratified by groups, under three different conditions of cervico-vaginal inflammation and infection: bacterial (Group A), viral (Group B) and fungal (Group C). The Pearson’s chi-square test reveals the significant association between the groups (A, B, and C) and the diagnostic outcomes for each test (Pap Test, HPV Test, Clinic Examination, Vaginal Ph, Vaginal swab, Colposcopy) with *p*-values less than 0.001 for all tests. The red letters (a, b, c) represent statistical significance at baseline (T0), while the blue letters (a, b, c) represent statistical significance at follow-up (T1). An equal subscript letter indicates a non-significant difference between groups.

	Group A *n* = 130		*p*-Value (T1 vs. T0)	Group B *n* = 117		*p*-Value (T1 vs. T0)	Group C *n* = 151		*p*-Value (T1 vs. T0)
	T0 *n* (%)	T1 *n* (%)		T0 *n* (%)	T1 *n* (%)		T0 *n* (%)	T1 *n* (%)	
**Pap Test**			<0.001			<0.001			<0.001
Positive	115 (89.2) a	15 (11.5) a		87 (74.4) b	33 (28.2) b		111 (73.5) b	30 (19.9) a,b	
Negative	14 (10.8) a	73 (56.2) a		26 (22.2) b	76 (65.0) b		40 (26.5) b	96 (63.6) a,b	
N/A	0	42 (32.3)		4 (3.4)	8 (6.8)		0	25 (16.6)	
**HPV test**			NS			<0.001			NS
Positive	14 (11.5) a	3 (2.3) a		107 (91.5) b	42 (35.9) b		4 (2.0) c	3 (2.0) a	
Negative	81 (62.3) a	87 (66.9) a		1 (0.8) b	68 (58.1) b		91 (60.9) c	87 (57.6) a	
N/A	34 (26.2)	40 (30.8)		9 (7.7)	7 (6.0)		56 (37.1)	61 (40.4)	
**Clinical examination**			<0.001			<0.001			<0.001
Positive	99 (76.9) a	20 (15.4) a		51(42.7) b	3 (2.6) b		93 (62.2) c	8 (5.3) b	
Negative	12 (9.2) a	96 (73.8) a		65 (56.4) b	80 (68.4) b		52 (33.8) c	138 (91.4) b	
N/A	18 (13.8)	14 (10.8)		1 (0.9)	34 (29.1)		6 (4.0)	5 (3.3)	
**Vaginal Ph**			<0.001			NS			<0.001
Altered	110 (85.4) a	20 (15.4) a		18 (14.5) b	17 (14.5) a		5 (3.3) c	25 (16.6) a	
Not altered	7 (5.4) a	82 (63.1) a		72 (62.4) b	95 (81.2) a		139 (92.1) c	114 (75.5) a	
N/A	12 (9.2)	28 (21.5)		27 (23.1)	5 (4.3)		7 (4.6)	12 (7.9)	
**Vaginal Swab**			<0.001			NS			<0.001
Positive	76 (58.5) a	15 (11.5) a		5 (4.3) b	17 (14.5) a		77 (51.0) a	40 (26.5) b	
Negative	2 (1.5) a	100 (76.9) a		14 (12.0) b	86 (73.5) a		8 (5.3) a	97 (64.2) b	
N/A	52 (40.0)	15 (11.5)		98 (83.8)	14 (12.0)		76 (43.7)	14 (9.3)	
**Colposcopy**			<0.001			<0.001			<0.001
Positive	89 (68.5) a	26 (20.0) a		107 (91.5) a	30 (25.6) a		123 (81.5) a	39 (25.8) a	
Negative	8 (6.2) a	56 (43.1) a		4 (3.4) a	73 (62.4) a		8 (5.3) a	88 (58.3) a	
NA	33 (25.4)	48 (36.9)		6 (5.1)	14 (12)		20 (13.2)	24 (15.9)	

## Data Availability

Data available on request due to privacy restrictions

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
