# Peer review of "Collagen-Based Ovule Therapy Reduces Inflammation and Improve Cervical Epithelialization in Patients with Fungal, Viral, and Bacterial Cervico-Vaginitis"

_medicina, 2023, doi:10.3390/medicina59081490_

Round 1
Reviewer 1 Report
1) Overall, the text is bad-written and requires extensive English correction (several phrases are difficult to comprehend). Here are some aspects the authors must address:
The article's title is too generic and does not reflect the study's results. Authors must change it in an appropriate way.
2) how the sample size was calculated?
Page 2, line 49- please justify the sentence.
Page 2, line 53- please justify the sentence.
Page 3, line 43 - please justify the sentence.
Page 3, line 50 - please justify the sentence.
Page 4, line 47- please justify the sentence.
Page 4, line 48- please justify the sentence.
Page 5, line 6- please justify the sentence.
Reference 3 - Please check the reference.
3) Elaborate your discussion
4) Author needs to reframe the title
5) Author needs to reframe the abstract with a complete description of the results and to mention the odds ratio of risk-associated study in the abstract section.
6) Reframe the rationale and objective of the study.
7) In the results section authors have not mentioned the significance level
8) The manuscript needs proofreading and correction of editing errors.
9) The results presented in this section of the manuscript are not clearly described.

Author Response
Dear Reviewer,
Thank you for your thorough review and constructive feedback. We appreciate your valuable insights, which will undoubtedly help us improve the clarity and quality of our manuscript. Here are our responses to each of the points you raised:
1) The article's title is too generic and does not reflect the study's results. Authors must change it in an appropriate way.
- We acknowledge your concern about the title and agree that it should accurately reflect the study's results. The title has been changed to provide greater clarity:
“Collagen-Based Ovule Therapy reduces inflammation and improve cervical epithelialization in Patients with Fungal, Viral, and Bacterial Cervico-Vaginitis.”
2) how the sample size was calculated?
- We apologize for the lack of clarity in some sentences. We have provided a detailed explanation in the text (Data analysis section) for how the sample size was calculated: “The power calculation for this retrospective observational study was conducted to determine the sample size required to detect meaningful differences in the effectiveness of Plurigin Ovules therapy across different types of cervico-vaginal infections. The parameters considered for the power calculation included an expected effect size based on preliminary data, a significance level (alpha) of 0.05, and a desired power of 0.80. Given the anticipated variability in treatment response among bacterial, viral, and fungal cervico-vaginitis cases, a sample size of approximately 500 participants was deemed necessary. This sample size ensured that the study had sufficient statistical power to detect significant differences in therapeutic outcomes, allowing for a comprehensive evaluation of the therapy's effectiveness in promoting cervical homeostasis and reducing infection recurrence.”
Page 2, line 49- please justify the sentence.
Page 2, line 53- please justify the sentence.
Page 3, line 43 - please justify the sentence.
Page 3, line 50 - please justify the sentence.
Page 4, line 47- please justify the sentence.
Page 4, line 48- please justify the sentence.
Page 5, line 6- please justify the sentence.
- All the above sentences have been justified.
Reference 3 - Please check the reference.
- Reference number 3 has been checked and corrected.
3) Elaborate your discussion
- We understand the importance of providing a more comprehensive discussion. The discussion has been elaborated to provide a deeper analysis of our results and their implications.
4) Author needs to reframe the title
- The title has been reframed: “Collagen-Based Ovule Therapy reduces inflammation and improve cervical epithelialization in Patients with Fungal, Viral, and Bacterial Cervico-Vaginitis.”.
5) Author needs to reframe the abstract with a complete description of the results and to mention the odds ratio of risk-associated study in the abstract section
- We revised the abstract to provide a complete description of the results, including mentioning the odds ratio of risk-associated study, as you suggested.
6) Reframe the rationale and objective of the study.
- A revision has been made to the introduction section in order to clarify the rationale and objective of the study
7) In the results section authors have not mentioned the significance level
- Significance levels have been added in the results section.
8) The manuscript needs proofreading and correction of editing errors.
- Proofreading and corrections have been made to the manuscript
9) The results presented in this section of the manuscript are not clearly described.
- We apologize for any confusion caused by unclear descriptions in the results section. The results have been rewritten to provide greater clarity. In addition, we have included a table with the calculated odds ratio and a figure (Figure1) that shows differences in colposcopy findings between the baseline and follow-up.
Once again, we appreciate your detailed feedback and your commitment to enhancing the quality of our manuscript. Your guidance will undoubtedly contribute to making our research more accessible and understandable to a wider audience.
Reviewer 2 Report
The retrospective study is generally reasonably designed, but the depth and workload of the study seem to be seriously inadequate, and of course there are some problems that need to be explained or solved. 1. Acute cervico-vaginitis is not mentioned in exclusion criteria of the study. If it is acute cervico-neginitis with more severe symptoms, we only carry out the treatment of supplementing vaginal lactic acid bacteria, can we guarantee a cure? What is the basis for treatment?2. What method is used for HPV detection? HPV typing? HC-2? HPV E6E7 mRNA? 3. What specific tests are included in the clinical examination in Table 3? How to tell if the result is positive or negative? 4. It is recommended to supplement 3 groups of colposcopic images before and after treatment.Author Response
The retrospective study is generally reasonably designed, but the depth and workload of the study seem to be seriously inadequate, and of course there are some problems that need to be explained or solved. 1. Acute cervico-vaginitis is not mentioned in exclusion criteria of the study. If it is acute cervico-neginitis with more severe symptoms, we only carry out the treatment of supplementing vaginal lactic acid bacteria, can we guarantee a cure? What is the basis for treatment?2. What method is used for HPV detection? HPV typing? HC-2? HPV E6E7 mRNA? 3. What specific tests are included in the clinical examination in Table 3? How to tell if the result is positive or negative? 4. It is recommended to supplement 3 groups of colposcopic images before and after treatment.
- Thank you for your insightful review of our study. We appreciate your feedback and would like to address your concerns and suggestions:
• Exclusion Criteria and Acute Cervico-vaginitis: We acknowledge the importance of clearly defining our exclusion criteria to ensure the accuracy and reliability of our participant selection process. As patients with acute cervico-vaginitis have not been included, explicit criteria for excluding patients with acute cervico-vaginitis in our study's exclusion criteria section have been added. The new sentence in exclusion criteria section is: “Individuals with acute cervico-vaginitis, characterized by severe inflammation of the cervix and vagina”.
• What is the basis for treatment: In our study we chose to use Plurigin Ovules because they content collagen, present in swine gelatine. The aim of the study was to investigate the potential effects of collagen on cervico-vaginal infections. Collagen, a key structural protein, is known for its role in maintaining tissue integrity and promoting healing. The presence of lactic acid in the formulation could further enhance the potential therapeutic effects and provide a comprehensive approach to addressing cervico-vaginal health.
• HPV Detection Method: We apologize for not providing sufficient information about the HPV detection method used in our study. We employed HPV DNA testing using the HC-2 assay for detecting the presence of HPV. We appreciate your input and we now included this information in the methodology section of our study.
• What specific tests are included in the clinical examination in Table 3? How to tell if the result is positive or negative?
The clinical assessment included the evaluation of leukorrhea, distinguished by foamy, dense, and yellowish discharge, as well as vulvovaginal erythema and edema. These evaluations were detailed in the "speculum examination" subsection of the Material and Methods (Data Collection) section. For improved clarity, these assessments were subsequently renamed as "clinical examination." If signs of erythema and edema are observed, the results are considered positive.
• Colposcopy Images: Your suggestion to include colposcopy images before and after treatment is valuable. We agree that visual evidence would provide enhanced insights into the effectiveness of the treatment. We added colposcopy images from three distinct groups (Figure 1), illustrating the baseline and post-treatment stages.
Reviewer 3 Report
Dear Authors,
I read your manuscript and I appreciate your work. I have some recommendations that can improve your article.
The abstract and introduction are adequate.
The methodology rises me an issue. Why did you chose Prurigin ovules? Is there any pharmaceutical involvement? Can you explain that issue? Why do not you use just the term of hialuronic ovules?
The result section is detailed and sustained by multiple data.
The discussion section can be improved with one or two paragraphs about other type of hyaluronic ovules.
Author Response
- Dear Reviewer,
Thank you for taking the time to review our manuscript. We appreciate your feedback and recommendations for improving the article. We would like to address the points you've raised:
Regarding your question about the choice of Plurigin ovules, we want to clarify that the basis of the treatment in our study is not hyaluronic acid but rather a collagen-based therapy. The ovules used in the study primarily contain collagen derived from swine gelatin, which is the fundamental component of the treatment. The reason we referred to it as "collagen-based therapy" is because collagen plays a crucial role in the potential therapeutic effects we aimed to investigate.
As for pharmaceutical involvement, we want to assure you that our study was conducted independently without any pharmaceutical company's direct involvement. The choice of using Plurigin ovules was based on their collagen content and the potential benefits of collagen-based therapy for cervico-vaginal infections. We aimed to explore the effects of collagen, specifically in the context of cervico-vaginal health. As of our knowledge, Plurigin Ovules are the only one present in the market that contain collagen.
In response to your suggestion about using the term "hyaluronic ovules," we acknowledge your point. However, since our study's primary emphasis was on the collagen component and its potential therapeutic impact, we believe that referring to the treatment as "Plurigin ovules" or “collagen-based therapy” better reflects the essence of our investigation.
We appreciate your positive feedback on the results section and are glad to hear that the data presentation is detailed and supported by multiple pieces of information.
Regarding your suggestion for including a discussion of other types of hyaluronic ovules, we believe that focusing on collagen-based therapy was our study's distinctive aspect. While other hyaluronic acid-containing ovules might have potential benefits, our study aimed to specifically explore the effects of collagen in cervico-vaginal infections. However, we do acknowledge the importance of discussing other ovule-based therapeutic approaches. In the discussion, we added a commentary on these approaches.
Once again, we thank you for your thoughtful review and valuable feedback. Your insights will undoubtedly contribute to improving the clarity and comprehensiveness of our manuscript.
Round 2
Reviewer 2 Report
We carefully reviewed the revised manuscript and the questions we raised, the revised version was well explained or resolved. We suggest that it is acceptable for publication.
Minor editing of English language required.